# The Genus *Sagina* (Caryophyllaceae) in Italy: Nomenclatural Remarks

**DOI:** 10.3390/plants12173169

**Published:** 2023-09-04

**Authors:** Duilio Iamonico, Laura Guglielmone, Emanuele Del Guacchio

**Affiliations:** 1Department of Environmental Biology, University of Rome Sapienza, Piazzale Aldo Moro 5, 00185 Rome, Italy; duilio.iamonico@uniroma1.it; 2Herbarium TO, Department of Life Sciences and Systems Biology, University of Turin, 10125 Torino, Italy; laura.guglielmone@unito.it; 3Department of Biology, University of Naples “Federico II”, Botanical Garden, via Foria 223, 80139 Naples, Italy

**Keywords:** *Alsinoideae*, heterotypic synonym, nomenclature, *Sagineae*, taxonomy, typification

## Abstract

A contribution to the nomenclature of the genus *Sagina* is presented. The following 10 taxa are recognized as being part of the Italian flora: *S. alexandrae*, *S. apetala*, *S. glabra*, *S. maritima*, *S. micropetala*, *S. nodosa*, *S. pilifera*, *S. procumbens*, *S. revelierei*, and *S. saginoides* subsp. *saginoides*. The names *S. apetala* var. *decumbens* (=*S. apetala* subsp. *apetala*), *S. bryoides* (=*S. procumbens*), *S. patula* (=*S. apetala* subsp. *apetala*), *S. revelierei*, *Spergula glabra* (=*S. glabra*), *Spergula pilifera* (=*S. pilifera*), and *Spergella subulata* var. *macrocarpa* (=*S. saginoides* subsp. *saginoides*) are here typified. Specimens deposited at B-W, C, E, and LY, and illustrations by Reichenbach were considered for the typifications. Specifically, two Reichenbach’s illustrations are chosen for *S. bryoides* and *S. saginoides* var. *macrocarpa*. A specimen at B-W is designated as the lectotype of *S. glabra*. Two specimens at C and G are designated as the lectotypes of *S. apetala* var. *decumbens* and *S. revelierei*, respectively. A specimen at LY is designated for *S. patula*. As we did not find original material, a neotype at G is designated for *S. pilifera*.

## 1. Introduction

*Sagina* L. (Sagineae J.Presl, Caryophyllaceae Juss.) is a genus of about 30 species mostly occurring in the North–temperate and arctic regions (center of diversity in Europe), while a few taxa can be found in tropical mountains [1]. Species of *Sagina* are small, erect, or procumbent (sometimes caespitose or cushion forming), as well as annual or perennial herbs; most species have white petals. A detailed comprehensive monography of the whole genus is still lacking, but molecular data demonstrated that *Sagina* is monophyletic [2,3].

The flora of Italy includes 11 *Sagina* taxa (all native), two of which, namely *S. pilifera* (DC.) Fenzl and *S. revelierei*, Jord. & Forr., are endemic [4].

As part of ongoing studies of Italian Caryophyllaceae [5,6,7,8,9,10,11,12,13,14,15], in this study, we present a contribution to the knowledge of *Sagina* in Italy by answering open nomenclatural questions and being preliminary to the taxonomic revision of the genus at the national level.

## 2. Results and Discussion

### 2.1. Publication and Typification of Names

#### 2.1.1. *Sagina apetala* var. *decumbens*

Hornemann [16] (p. 3) proposed two varieties of *Sagina apetala*, i.e., *S. apetala* var. *erecta* and *S. apetala* var. *decumbens*, and associated them with an illustration (“Table MMCII”, available at http://bibdigital.rjb.csic.es/ing/Libro.php?Libro=4640&Pagina=160, accessed on 24 August 2023 This illustration represents two complete plants (one placed on the top of the plate and the other placed at the center of the plate), as well as a series of eight detailed drawings (lower part of the plate), showing the particulars of leaves, gynoecium and androecium, a first fruit with adpressed sepals, a second fruit with patent sepals, and a seed. According to the protologue by Hornemann (1834: 3), the upper plant drawing illustrates *S. apetala* var. *erecta* (“Fig. [=Figura] superior var. *erecta*”), whereas the complete figure corresponds to *S. apetala* var. *decumbens* (“Fig. [=Figura] infer. var. *decumbens*”). Although no diagnosis was provided for the two varieties, the names can be considered as validly published based onto Art. 38.9 of the ICN.

Dillenberger and Kadereit [17] (p. 20) lectotypified the name *Sagina apetala* var. *erecta* using a specimen deposited at C (C10024083), whereas no definitive conclusion was given for *S. apetala* var. *decumbens*. These authors [17] stated that the “protologue [of *S. apetala* varieties by Hornemann] does not provide separate information for *S. apetala* var. *erecta* and var. *decumbens*. Specimen No. 3 on the sheet consists of three large plants all belonging to *S. micropetala* and has the correct location, collector and date. Specimen No. 1 seems to include *S. apetala* var. *decumbens*, No. 2 lacks a date, and label information of No. 4 does not fit the protologue”. Furthermore, in the text, Dillenberger and Kadereit [17] (p. 15) reported in the text that: “The growth form of var. *decumbens* clearly excludes it from *S. apetala* and *S. micropetala* so that var. *decumbens* most likely represents *S. procumbens*”. Therefore, as a whole, the authors [17] had doubts about the identification of Hornemann’s *S. apetala* var. *decumbens*. We carefully checked both C10024083 [17] (p. 19, Figure 2) and the treatment of *Sagina* by Hornemann [p. 3 and Tables MMCII (*Sagina apetala*), MMCIII (*Sagina procumbens* L.), and MMCIV (*Sagina maritima*)] [18]. Specimen No. 1 in C10024083 consists of three complete plants (with roots, leaves, sepals, and fruits) and it is associated with the following label (bottom-right corner of the sheet): “1|*Sagina apetala* L. var. *depressa* Schultz|... bei Ratzeburg | leg. Nolte”. Two of these three plants (the first plants on the left) of specimen No. 1 in C10024083 are clearly prostrate and, therefore, match Hormenan’s concept of *S. apetala* var. *decumbens* [16] (Table MMCII–central illustration). Although the year of collection is not specified on this label, the collector and the locality are compatible with the protologue, as Ratzeburg is the capital of the district Herzogtum Lauenburg (Lower Saxony, Germany). Therefore, it is very likely that the Danish botanist used these plants to describe *S. apetala* var. *decumbens*, as well as using the plants of specimen No. 3 to describe *S. apetala* var. *erecta* [17]. Thus, we designate this latter as the lectotype of *Sagina apetala* var. *decumbens*.

Concerning the identity of *Sagina apetala* var. *decumbens*, we firstly noted that most of the authors cited by [17] (p. 3) in synonymy [18,19,20,21,22] referred to the taxon currently known as *S. apetala* subsp. *apetala* (excluding Nolte and Weber [23] (p. 119), who treated *Sagina maritima* Don). On the other hand, as mentioned above, Dillenberger and Kadereit [17] (p. 15) supposed that Hornemann’s *S. apetala* var. *decumbens* could be *S. procumbens*. This latter species is the only perennial *Sagina* member among those species with tetramerous flowers (see, e.g., [17,24]). Both the lectotype chosen and the illustrations by Hornemann [16] (Table XXCII) clearly show annual plants. The detailed drawing at the bottom of the plate (the second one from the right) that shows fruits with patent and obtuse sepals is referred to as *S. apetala* var. *erecta* (=*S. micropetala*, according to [17]). The other fruit drawn, which is subtended by adpressed and acute sepals (the third drawing from the right on the bottom of Table MMCII), clearly is *S. apetala* var. *decumbens* (=var. *apetala*).

#### 2.1.2. *Sagina bryoides*

Reichenbach [25] (p. 793) published the name *Sagina bryoides*, providing a diagnosis and provenance (“*Tyrol, im Bassin Lyn bei Steeg*”), gathering details (“*Frölich* [Froelich]”, “31 Aug. 1811”), and citing an illustration of his own [25]: “*Rchb.* pl. crit. XI. ic. …”. Evidently, this illustration, as well as that of the subsequent *Sagina ciliata* Fr. [25], was planned to be published within the 19th *centuria* in *Iconographia botanica seu plantae criticae*. However, as clarified by Stafleu and Cowan [26], the series stopped after the 10th volume in either 1832 or 1834, and the remaining centuries of plates were incorporated into the first volumes of *Icones florae germanicae et helveticae*. In particular, the first volume would count as being published in the century of *Iconographia botanica*, but it is entirely devoted to monocots and, therefore, titled “*Agrostographia germanica*” [26]. The illustration of interest was only published in 1841 or 1842 within the fifth volume of *Icones florae germanicae et helveticae*, namely as the No. 4955 of the plate CC [27]. We do not know if the plate had previously been realized when Reichenbach published the protologue [25]; therefore, even if this is possible, to the best of our knowledge, it cannot be considered as original material. Nevertheless, the syntype cited by Reichenbach [25], i.e., a specimen collected by Froelich on 31 August 1811 in Tyrol, as well as any further specimen linked to Reichenbach, was not traced. According to Stafleu and Cowan [28] (p. 893), the hosting place of Froelich’s Herbarium is unknown. Therefore, the illustration cited by Reichenbach [25], albeit published only nine years later, remains the only known material definitively linked to him, and future studies might consider it to be original material. Therefore, we prefer to designate it as the neotype of the name *Sagina bryoides*. On the basis of the current concept in *Sagina* (see e.g., [24]), *S. bryoides*, and the chosen neotype, it can be referred to *S. procumbens*.

#### 2.1.3. *Sagina revelierei*

Jordan and Fourreau [29] (p. 11) published the name *Sagina revelierei* (as “*revelieri*”) to honor the first collector Eugène Revelière; this name does not have a well-established Latin form and, therefore, must corrected according to Arts. 60.1 and 60.8 (a) of the ICN. Jordan and Fourreau [29] provided a detailed diagnosis and the provenance (“Hab. in montibus Corsicae: *Quenza*, ex dom. E. Revelière”); they also provided a morphological comparison between *S. linnaei* C.Presl and *S. subulata*, taking the work of Presl into account [30] (p. 14), which listed the valid *Spergula saginoides* L. as a synonym of *S. linnaei*. As a consequence, *S. linnaei* is a superfluous and illegitimate name under the Arts. 52.1 and 52.2 of the ICN.

Following Stafleu and Cowan [26] (p. 746), we searched Revelière’s specimens and traced two sheets at LY [LY0826457 and LY0826458 (at https://explore.recolnat.org/occurrence/989AB9E6EAB340EFA3629CD7DB0C398C, accessed 19 July 2023)]. These specimens were collected by Reveliére in 1863 at Quenza, and Jordan in 1865 at the Garden of Villeurbanne (plants grown from seedling cultivated in 1864 and coming from Quenza), respectively. Both specimens, which can be considered to be part of the original material for the name *Sagina revelierei*, morphologically match the protologue and the current concept of the species [24]. We designate the specimen LY0826457, which was directly collected by Reveliére, as the lectotype of *Sagina revelierei*.

#### 2.1.4. *Spergula glabra*

This name, which is currently reported to be a synonym of *Sagina glabra* (Willd.) Fenzl, was published by Willdenow [31] (p. 821). The protologue consists of a short diagnosis (“S. [Spergula] foliis oppositis fasciculatis filiformibus glabris, floribus decandris, petalis calyce majoribus”), the illustration of “Spergula saginoides Allion. pedem. n. 1735. t. 64. f. 1” (a misapplication by Allioni of the Linnean name *S. saginoides*), and the provenance (“*Habitat in pascuis umbrosis alpium* Valdensium”). Willdenow’s name can be typified using material deposited in his collection at B or Allioni’s iconography from *Flora Pedemontana.*

There is one sheet at B (B-W09052; https://ww2.bgbm.org/Herbarium/specimen.cfm?Barcode=BW09052010, accessed on 10 August 2023) that bears two plants, namely Willdenow’s scripts “*Sp.* [Spergula] *glabra*” (top right corner of the specimen) and “*W.* [Willdenow]” (bottom-right-corner), which is a label (front of the sheet) reporting the diagnosis (“*Decandria Pentagina*|*Spergula glabra foliis oppositis fasciculatis glabris, floribus decandris petalis calyce majoribus*|*Sp. Pl. 2. P 821*”), and a second label (on verso of the sheet) with the following script: “*Spergula saginoides|H. Ped.* [Hortus Pedemontanus] *vivam, diversa a planta Linnaei et Halleri (Bellardi)*” (C.A.L. Bellardi was a pupil and close collaborator of Allioni, and Allioni often examined Bellardi’s material). We consider this B sheet to be part of the original material used by Willdenow [31] to describe *Spergula glabra*.

Between the two elements suitable for the typification of the name *Spergula glabra*, we decide to designate the specimen at B as the lectotype, since it is a better choice regarding the illustrations [32] (pp. 21−22).

#### 2.1.5. *Spergula pilifera*

Lamarck and Candolle [33] (p. 774) provided a detailed diagnosis of the name *Spergula pilifera*, using data about its provenance and collectors (“Elle croît sur les hautes montagnes de l’isle de Corse, d’où M. Robert en a envoyé des échantillons que je décris dans l’herbier de M. Clarion” = it grows on the high mountains of the island of Corsica, where Mr. Robert sent samples of it, which are described in Mr. Clarion’s herbarium); this latter citation is a sintype according to Art. 9.6 of the ICN.

According to Stafleu and Cowan [28] (p. 506), Clarion’s collection is preserved at P-JU (Jussieu Herbarium). Unfortunately, among the 17 specimens preserved at P (see https://science.mnhn.fr/institution/mnhn/collection/p/item/list?scientificName=Spergula+pilifera, accessed on 10 August 2023), none are part of the original material (it was either not collected by Clarion or not noted before 1805). Also, C. Aupic (pers. comm.) confirmed that no Clarion’s specimen of *Sagina pilifera* is deposited at P. As a consequence, a neotypification is required by the Art. 9.8 of the ICN. We traced a specimen at G (G00212132), which is represented by one plant collected by Robert in 1808 in Corsica. This exsiccatum is, therefore, a good candidate to be a neotype for the name *Spergula pilifera*, as it morphologically corresponds both to the protologue and to the current concept of *Sagina pilifera* [26].

#### 2.1.6. *Spergella subulata* var. *macrocarpa*

Reichenbach [27] (p. 26) proposed the name *Sagina saginoides* var. *macrocarpa* to identify plants occurring in South Tyrol (Northern Italy, Trentino-Alto Adige region). He reported “Spergula saginoides Pollini Fl. Veron. II. T. I. f. 2”, which corresponds to *Spergula saginoides sensu* Pollini [34] (p. 75). Reichenbach [27] provided an illustration that is original material for var. *macrocarpa*. According to Crow [35] (p. 42), Reichenbach’s specimens, which would be useful for typification purposes, were destroyed. As a consequence, Reichenbach’s image appears to be the only extant material, and it is here designated as the lectotype of the name *Spergella subulata* var. *macrocarpa*. We agree with Crow [35] (p. 34, 42) in terms of synonymizing Reichenbach’s name with *S. saginoides*.

### 2.2. Conspectus of the Genus Sagina in Italy

We here present a nomenclatural conspectus of the following 10 *Sagina* taxa occurring in Italy:**1**.**Sagina alexandrae:** Iamonico, Phytotaxa 282(2): 164. 2016, nom. nov. pro Sagina subulata (Sw.) Presl, Fl. Sic. 1: 158. 1826, non d’Urv. ≡ Spergula subulata Swartz, Kongl. Vetensk. Acad. Nya Handl. sér. 2, 10(1): 45. 1789 ≡ Alsine subulata (Sw.) Jessen, Deut. Excurs.-Fl.: 286. 1879 ≡ Moehringia subulata (Sw.) Clairv., Man. Herbor. Suisse: 150. 1811 ≡ Phaloe subulata (Sw.) Dumort., Fl. Belg.: 110. 1827 ≡ Sagina linnaei var. subulata (Sw.) Fiori, Fl. Italia [Fiori, Béguinot, and Paoletti] 1(2): 340. 1898 ≡ S. saginoides var. subulata (Sw.) Fiori, Nuov. Fl. Italia 1: 456. 1923.

Lectotype [designated by Crow [35] (p. 51)]: [Sweden] Halland, *Osbeck s.n.* (BM; possible isolectotype SBT14207, the image of the isolectotype is available at http://plants.jstor.org/stable/10.5555/al.ap.specimen.sbt14207, accessed 24 August 2023).

**Note:** POWO [36] reported that *Sagina alexandrae* is a synonym of *S. hawaiesis* Pax. However, the question arising from the synonymization of these two names is quite complicated, and Iamonico and Dillenberger (in prep.) are studying it from both nomenclatural and molecular points of view.

**2**.***Sagina apetala*** Ard., Animadv. Bot. Spec. Alt.: 22–23. 1764 subsp. ***apetala*** ≡ *Sagina erecta* L., Sp. Pl. 1: 128. 1753 var. *apetala* (L.) Lam., Fl. Fr. 3: 9. 1778 ≡ *Alsine apetala* (Ard.) Jess., Deutsche Excurs.-Fl.: 287. 1879 ≡ *Alsinella apetala* (Ard.) Krause, Deutschl Fl. (Sturm), ed. 2. 5: 38. 1901.

Lectotype [designated by Crow [35] (p. 73)]: Herb. Linnaeus No. 177.2 (LINN!, the image of the lectotype is available at http://linnean-online.org/2016/, accessed on 10 August 2023).

=*Sagina apetala* var. *decumbens* Hornemann, Fl. Dan. [Oeder] 12(36): 3. 1834.

Lectotype (**designated here**): GERMANY. Bei Ratzeburg, s.d., *Nolte s.n.* [C10024083 specimen No. 1 on the top of the sheet; image in [17] (Figure 2 at p. 15)].

=*Sagina ciliata* Fr., Utkast Sv. Fl., ed. 3: 713. 1816.

Lectotype (designated by Thullin [37] (p. 222)): [Sweden] Skåne, “in arvis et inter segetes ad Nibelöf inter Ystad et Trelleborg”, *s.d.*, *Fries s.n.* (UPS-V-1051094!).

=*Sagina stricta* Fr., Novit. Fl. Suec. Mant. 3: 122. 1817.

Lectotype [designated by Thullin [37] (p. 222)]: [Sweden] Skåne, “Cimbritshamn” [Simrishamn], 7 July 1817, *Fries s.n.* (UPS-V-1051091!).

=*Sagina apetala* var. *barbata* Fenzl in C. F. von Ledebour, Fl. Ross. 1: 338. 1842.

Lectotype: Not designated.

=*Sagina patula* Jord., Observ. Pl. Nouv. 1: 23. 1846.

Lectotype (designated by Dillenberger and Kadereit [17] (p. 17): [France] Dans les terres argileuses a graminacean / Rhône, 02 June 1845, *Jordan s*.*n*. (LY0826462!, the image of the lectotype is available at https://explore.recolnat.org/search/botanique/simplequery=LY0826462, accessed on 10 August 2023).

=*Sagina melitensis* Giulia ex Duthie, J. Bot. 13: 37. 1875.

Lectotype [designated by Crow [35] (p. 73)]: [Malta] Corradino, 13 March 1874, *Dulthie s.n.* (K000723212!, the image of the lectotype is available at https://plants.jstor.org/stable/viewer/10.5555/al.ap.specimen.k000723212?loggedin=true, accessed in 24 August 2023).

**Notes on *Sagina apetala* var. *barbata*:** Crow [35] (p. 73) proposed a specimen at LE (LE00012064, the image of the specimen is available at https://plants.jstor.org/stable/viewer/10.5555/al.ap.specimen.le00012064, accessed on 10 August 2023) collected by E. Fenzl (unkown locality of collection) as the lectotype of the name (it was reported as “Lectotype, LE!”). However, Fenzl [38] (p. 338) reported two syntypes in the protologue: “Livonia! (C. A. Meyer)” and “Podolia occidentali! (Besser)”. We noted that Fenzl [38] (p. 338) did not link these two syntypes to his two described varieties (var. *barbata* and var. *imberbis*). Regardless, according to Arts. 9.6 and 9.12 of the ICN, “In lectotype designation, an isotype must be chosen if such exists, or otherwise a syntype or isosyntype if such exists”. Thus, LE00012064, which is not a syntype, cannot be considered to be the lectotype of *Sagina apetala* Ard. var. *barbata*. This LE specimen could be considered to be a neotype according to Art. 9.10 of the ICN, albeit only if no original material exists (Arts. 9.4 and 9.8 of the ICN). Further investigation of these Fenzl’s names is needed.

**Notes on *Sagina melitensis*:** Crow [35] (p. 73) correctly lectotypified the name *Sagina melitensis* by reporting “Lectotype¸K!”. In fact, According to Art. 7.11 of the ICN, the phrase “designated here” was not necessary up to 1 January 2001.

**3**.***Sagina glabra*** (Willd.) Fenzl, Vers. Darstell. Alsin.: 57. 1833 ≡ *Spergula glabra* Willd., Sp. Pl., ed. 4. 2: 821. 1799 ≡ *Spergella glabra* (Willd.) Rchb., Fl. Germ. Excurs. 2: 794. 1832.

Lectotype (**designated here**): [Italy] *H. Ped.* [=Hortus Pedemontanus, i.e., a cultivated plant probably collected in Valdesium (Piemonte region, Northwestern Italy)], s.d., *P. Bellardi[?] s*.*n*. (B-W09052-010!, the image of the lectotype is available at http://ww2.bgbm.org/herbarium/specimen.cfm?SpecimenPK=137061&idThumb=342876&SpecimenSequenz=1&loan=0, accessed on 10 August 2023).

−“*Spergula saginoides*” *sensu* Allioni, Fl. Pedem. 2: 118. 1785.

**4**.***Sagina maritima*** Don, Herb. Brit. 7: n.° 155. 1806 ≡ *Alsine maritima* (Don) Jess., Deut. Excurs.-Fl.: 287. 1879 ≡ *Alsinella maritima* Bubani, Fl. Pyren. 3: 56. 1901.

Lectotype [designated by Dillenberger and Kadereit [17] (p. 18)]: [United Kingdom] Scotland, s.d., *Don s.n.* (E00455340!, the image of the lectotype is available at http://elmer.rbge.org.uk/bgbase/vherb/bgbasevherb.php?cfg=bgbase/vherb/zoom.cfg&filename=E00455340.zip&queryRow=4, accessed on 10 August 2023); isolectotype PH00022257, *fide* Dillenberger and Kadereit [17] (p. 18).

**5**.***Sagina micropetala*** Rauschert, Feddes Repert. 79: 413. 1969 = *Sagina apetala* var. *erecta* Hornem., Fl. Dan. [Oeder] 12(36): 3. 1834 ≡ *Sagina apetala* Ard. subsp. *erecta* (Hornem.) F.Herm., Fl. Deutschl. Fennoskand.: 182. 1912.

Lectotype [designated by Dillenberger and Kadereit [17] (p. 20)]: [Germany] Heiligenhaven, Aug 1825, *Nolte s.n.* (C10024083, specimen No. 3 on sheet).

**Note:** According to Art. 41.4 of the ICN, despite no direct or indirect reference to Hornemann’s var. *erecta* was reported, Hermann’s name [39] can be considered as a new combination at subspecies rank (the author reported “2 Unterarten:”, i.e., “2 subspecies”), not a new taxon as apparently it would seem. For the ascription by [40] of the var. *erecta* to Lamarck, see [17].

**6**.***Sagina nodosa*** (L.) Fenzl, Vers. Darstell. Alsin.: 18. 1833 ≡ *Spergula nodosa* L, Sp. Pl. 1: 440. 1753 ≡ *Alsine nodosa* (L.) Crantz, Inst. Rei Herb. 2: 408. 1766 ≡ *Moehringia nodosa* (L.) Clairv., Man. Herbor. Suisse: 150. 1811 ≡ *Arenaria nodosa* (L.) Wallr., Sched. Crit.: 200. 1822 ≡ *Alsinella nodosa* (L.) Bubani, Fl. Pyren. 3: 54. 1901.

Lectotype [designated by Crow [35] (p. 25)]: Herb. Linnaeus No. 604.4 (LINN!, the image of the lectotype is available at http://linnean-online.org/4282/, accessed on 10 August 2023).

−“*Alsine nodosa*” Krause, Deutschl. Fl. (Sturm), ed. 2. 5: 38. 1901, isonym (Art. 6 Note 2 of the ICN).

−“*Sagina nodosa*” Meyer, Elench.: 29. 1835, isonym (Art. 6 Note 2 of the ICN).

**7**.***Sagina pilifera*** (DC.) Fenzl, Vers. Darstell. Alsin.: 57. 1833 ≡ *Spergula pilifera* DC. In J.B.A.M.de Lamarck and A.P.de Candolle, Fl. Franç., ed. 3, 4: 774. 1805.

Neotype (**designated here**): [France] Corse, 1808, *Robert s.n.* (G00212132!, the image of the neotype is available at https://www.ville-ge.ch/musinfo/bd/cjb/chg/adetail.php?id=211326&lang=fr, accessed on 10 August 2023).

**8**.***Sagina procumbens*** L., Sp. Pl. 1: 128. 1753 subsp. ***procumbens***

Lectotype [designated by Jonsell and Jarvis in [41] (p. 83)]: [Icon] *Alsine pusilla graminea, flore tetrapetalo”* in Séguier (1745: 421, t. 5, f. 3, the image of the lectotype is available at http://bibdigital.rjb.csic.es/ing/Libro.php?Libro=4861&Pagina=495, accessed in August 2023).

=*Sagina bryoides* Froel. ex Rchb., Fl. Germ. Excurs. 2: 793. 1832 ≡ *Sagina procumbens* var. *typica* f. *bryoides* (Froel. ex Rchb.) Fiori, Fl. Ital. 1(2): 340. 1898 ≡ *Sagina procumbens* var. *Bryoides* (Froel. ex Rchb.) Fiori, Nuov. Fl. Italia 1: 455. 1923 ≡ *Sagina procumbens* subsp. *Bryoides* (Froel. ex Rchb.) Dostál, Folia Mus. Rerum Nat. Bohemiae Occid., Bot. 21: 4. 1984.

Neotype (**designated here**): [Icon] 4955. *Sagina bryoides* in Rchb., Icon. Fl. Germ. 5.: Table CC No. 4955. 1841–1842.

=*Sagina corsica* Jord., Observ. Pl. Nouv. 7: 15. 1849 ≡ *S. procumbens* subsp. *corsica* (Jord.) Rouy and Foucaud, Fl. France 3: 287. 1896 ≡ *S. procumbens* var. *corsica* (Sw.) Fiori, Nuov. Fl. Italia 1: 456. 1923.

Lectotype [designated by Crow [35] (p. 42)]: [France] Corse, Cagnone, July 1840, *Jordan s.n.* (P04023513!, the image of the lectotype is available at https://mediaphoto.mnhn.fr/media/1582102227767oEFHJ2NwcZYPwICc, accessed in August 2023).

=*Sagina muscosa* Jord., Mém. Acad. Sci. Lyon, Sect. Sci. 1: 32. 1852.

Lectotype [designated by Crow [35] (p. 42)]: [France] *Mt. Pilat (Loire)*, 1843, *Jordan 187* (MPU021802!, the image of the lectotype is available at https://plants.jstor.org/stable/viewer/10.5555/al.ap.specimen.mpu021802, accessed in August 2023).

**9**.***Sagina revelierei*** Jord. and Fourr., Brev. Pl. Nov. 1: 11. 1866 ≡ *Sagina subulata* subsp. *revelierei* (Jord. and Fourr.) Rouy and Fouc., Fl. France 3: 294. 1896 ≡ *Sagina saginoides* var. *revelierei* (Jord. and Fourr.) Fiori, Nuov. Fl. Italia 1: 156. 1923 ≡ *Sagina subulata* var. *revelierei* (Jord. and Fourr.) Fourn., Quatre Fl. France: 312. 1936.

Lectotype (**designated here**): [France] Cors, Quenza, 1863, *Reveliére 484* (LY0826457! image available at https://explore.recolnat.org/occurrence/132FBF97629B4B5FBF388D54D2CC6766, accessed in August 2023).

**10**.***Sagina saginoides*** (L.) H.Karst., Deut. Fl.: 539. 1882 subsp. ***saginoides***≡ *Spergula saginoides* L., Sp. Pl. 1: 441. 1753 ≡ *Alsine saginoides* (L.) Crantz, Inst. Rei Herb. 2: 408. 1766 ≡ *Alsinella saginoides* (L.) Green, Fl. Francisc.: 125. 1891 ≡ *Phaloe saginoides* (L.) Dumort., Fl. Belg.: 110. 1827 ≡ *Spergella saginoides* (L.) Rchb., Fl. Germ. Excurs. 2: 794. 1832 ≡ *Sagina linnaei* C.Presl, Reliq. Haenk. 2: 14. 1831, nom. superfl. et illeg. (Art. 53.3 of the ICN) ≡ *Alsine linnaei* Krause, Deutschl. Fl. (Sturm), ed. 2. 5: 35. 1901.

Lectotype [(designated by Crow [35] (p. 73)]: Herb. Linnaeus No. 604.06, *Gmelin s.n.* (LINN!, the image of the lectotype is available at http://linnean-online.org/4284/, accessed in August 2023).

=*Spergula micrantha* Bunge in C.F. von Ledebour, Fl. Altaic. 2: 183. 1830 ≡ *Sagina linnaei* var. *micrantha* (Bunge) Fenzl in C.F. von Ledebour, Fl. Ross. 1: 339. 1842 ≡ *Sagina micrantha* (Bunge) Fernald, Rhodora 27: 130. 1925.

Lectotype [designated by Crow [35] (p. 34)]: [Russia] Pr. [Prope] Barnaul, Tomskoi Sawod, s.d., *Bunge s.n.* (LE, *non vidi fide* Crow [35] (pp. 34, 42)].

=*Sagina saginoides* var. *macrocarpa* Rchb., Icon. Fl. Germ. Helv. 5: 26. 1841 ≡ *Sagina macrocarpa* (Rchb.) Maly, Enum. Pl. Phan. Austria: 243. 1848 ≡ *Sagina linnaei* var. *macrocarpa* (Rchb.) Beck, Fl. Niederosterreich: 358. 1890 ≡ *Sagina saginoides* subsp. *macrocarpa* (Rchb.) Soó, Acta Bot. Acad. Sci. Hung.: 177. 1973 ≡ *Sagina linnaei* var. *typica* f. *macrocarpa* (Rchb.) Fiori, Fl. Italia [Fiori, Béguinot and Paoletti] 1(2): 340. 1898 ≡ *Sagina saginoides* var. *macrocarpa* (Rchb.) Moss, J. Bot. 52: 60. 1914.

Lectotype (**designated here**): [Icon] *Sagina saginoides* var. (b) *macrocarpa* in Reichenbach (1841: Table CCII, the image of the lectotype is available at http://www.biodiversitylibrary.org/item/28648#page/199/mode/1up, accessed in August 2023).

−“*Sagina saginoides*” Britton, Mem. Torrey Bot. Club 5: 151. 1894, isonym (Art. 6 Note 2 of the ICN).

−“*Sagina saginoides* var. *macrocarpa*” Reichenbach, Icon. Fl. Germ. Helv. (H.G.L. Reichenbach) 5: 26. 1841, isonym (Art. 6 Note 2 of the ICN).

**Note:** Crow [35] (p. 34) and Tropicos [42] report that Beck’s combination is illegitimate. However, the name was correctly proposed by Beck (Art. 55.2 of the ICN).

## 3. Material and Methods

The present study was carried out through an extensive literature analysis and the examination of specimens kept at B, BM, C, E, G, K, L, LE, LY, MPU, NAP, P, PH, RO, TO, and UPS (herbarium codes were determined according to the *Index Herbariorum* [43]). The articles cited through the text follow the *Shenzhen Code* (hereafter “ICN”) [42]. Besides the accepted names and their basionyms, all of the homotypic synonyms were examined, and the most frequent heterotypic names.

## 4. Conclusions

On the basis of a detailed study of both the protologues and the original material regarding several names of *Sagina* and *Spergula*, we are able to clarify their identities via the designation of neo- or lectotypes. Most of these names are confirmed to be heterotypic synonyms. Concerning the homotypic synonyms, we noted that many genera were considered to identify the various taxa, e.g., *Alsine* L. (currently *Stellaria* L.), *Moehringia* L. (currently accepted), *Phaloe* Dumort. (currently *Sagina*), *Spergella* Rchb. (currently *Sagina*), *Spergula* L. (currently accepted), etc. Also, the various taxa were originally described or recombined during the time at all ranks, ranging from species to subcategories (subspecies, variety, and form). This proliferation of names and taxa reveals the high phenotypic variability in the genus *Sagina*, on one hand, as well as the confusion in their attribution to both the genus and the correct rank, on the other hand. The nomenclatural study here presented thus shows the importance of nomenclature in Systematic Botany, as it provides a base for future taxonomic investigations.

## Data Availability

Not applicable.

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
