# Peer review of "The Genus Sagina (Caryophyllaceae) in Italy: Nomenclatural Remarks"

_plants, 2023, doi:10.3390/plants12173169_

Round 1
Reviewer 1 Report
A few typos:
p.2. Fuhtermore should be Furthermore. theuy should be they.
p.5 needs should be is needed.
p.7 saginides should be saginoides.
many genera was should be many genera were considered
A few minor expressions in the English could be improved.
Author Response
We thank the reviewer and accepted all the suggestions by him.

Reviewer 2 Report
This is an interesting contribution focused on the taxonomy of the genus Sagina (Caryophyllaceae) in Italy.
I do have some comments and suggestions that could improve the manuscript (please, see the version after review).

Minor revision required.
Author Response
We are deeply indebted to the reviewer for his very valuable and patient work. We accepted almost all the suggestions by him as it can be seen in the punctual replies to his comments reported in the attached file. Unfortunately, I am not able to upload also the revised file here, but I reccomend to the Editor to make it visible to the reviewers.
